# Effect of Methyl Hydro-Silicone Oil Content and Aging Time on Compression Modulus and Breakdown Strength of Additional Liquid Silicone Rubber Gel

**DOI:** 10.3390/polym16060763

**Published:** 2024-03-10

**Authors:** Kun Wang, Yun Chen, Wei Yang, Bo Qiao, Jian Qiao, Jianfei He, Qinying Ning

**Affiliations:** 1Beijing Institute of Smart Energy, Beijing 102209, China; wangkun278@163.com (K.W.); chenyunsgri@163.com (Y.C.); qiaobo@bise.hrl.ac.cn (B.Q.); buaa_joe@126.com (J.Q.); m13718204573@163.com (J.H.); 2School of New Energy, North China Electric Power University, Beijing 102206, China

**Keywords:** vinyl silicone oil, methyl hydro-silicone oil, compression modulus, breakdown strength, thermal-oxidative aging

## Abstract

The performance of silicone rubber gel elastomers is affected by the composition and structure of the crosslinker. In this work, a two-component addition liquid silicone rubber gel material was developed, and the effects of the contents of two methyl hydro-silicone oils on the compression modulus and breakdown strength of the silicone rubber gel insulating material, as well as the performance change after hot air aging at different times (24 h, 48 h, 72 h, 96 h, 120 h, 144 h, 168 h), were studied. The results showed that the breakdown strength and compression modulus exhibited an upward trend with the increase in the hydrogen silicone oil content. The best performance was achieved in the silicone rubber gel with Si-H:Si-Vi = 1.4:1. Moreover, with the increase in aging time, the breakdown strength decreased and the compression modulus increased.

## 1. Introduction

As the core device in the field of power electronics, semiconductor power devices have a significant impact on their performance due to their packaging materials [1,2,3]. Additive liquid silicone rubber gel (ALSR) is the preferred material for semiconductor power device insulation and packaging due to its excellent chemical stability, high- and low-temperature resistance, and electrical insulation. ALSRs are prepared by a silicon–hydrogen addition reaction, an addition-crosslinking method that minimizes volume shrinkage and by-product formation in polycondensation reactions, eliminating the need for a complex and demanding process [4].

The matrix, crosslinker composition and structure, catalyst, and inhibitor content have a great influence on the gel’s performance. In the additional two-component liquid silicone rubber gel, the main substances of components A and B are methyl hydrogen silicone oil and methyl vinyl silicone oil, respectively, and the viscosity of the mixture is low before curing, which can result in vacuum encapsulation of complex geometries. Pt has been used as a catalyst to complete the silicon–hydrogen–vinyl addition reaction. After curing, the silicone elastomer is stable and has the advantages of being moisture-proof, dust-proof, anti-fouling, anti-impact, and anti-vibration, which can significantly improve the insulation performance of the device [5].

The maximum internal temperature of semiconductor chips in semiconductor power devices (IGBT) reaches as high as 175 °C, and long-term operation in a high-temperature environment will change the chemical structure and internal composition of silicone rubber gel for IGBT packaging, and these changes in material structure will lead to the degradation of their related properties, so the insulation properties of the silicone rubber gel will deteriorate to a certain extent [6,7,8]. La et al. [9] found that silicone rubber gel becomes hard and brittle after being subjected to heat for a long time at high temperatures (200 °C or higher), completely losing its elasticity and eventually cracking. The artificial accelerated thermal aging test was carried out on a silicone rubber gel specimen at 120 °C, which showed that the relative permittivity of the silicone rubber gel specimen after thermal aging decreased, the hardness of the material gradually increased, and the crosslinking density increased with the increase in thermal aging time.

Silicone rubber gel can maintain a solid shape, and due to the advantages of dimensional stability, small shrinkage, deep curing, low crosslinking density, more flexible molecular chains, and smaller installation pressure, the addition of liquid silicone rubber gel is more studied and applied in the packaging of semiconductor SiC power devices [10,11]. In the current research, there is a lack of systematic research on the influence of hydrogen silicone oil type and content and aging time on the properties of additional liquid silicone rubber gel materials. In this article, we investigate the effects of different component contents on the properties of silicone rubber gels. In this work, vinyl silicone oil (ViSil-100) and two methyl hydro-silicone oils (HSil-65 and HSil-100) were used to study the effects of hydrogen silicone oil content and aging time on the compression modulus and breakdown characteristics of the silicone rubber gel insulating material. The results showed that with an increase in the methyl hydro-silicone oil content, the breakdown strength and compression modulus showed an upward trend. When Si-H:Si-Vi = 1.4:1, the performance of the insulating material was the best. With the increase in aging time, the sample defects increased, the breakdown strength decreased, and the compression modulus increased due to the increase in crosslinking density.

## 2. Materials and Methods

### 2.1. Materials

Vinyl silicone oil ViSil-100 (RH-Vi1321, vinyl content of 1.07 wt%, viscosity of 100 mPa·s), methyl hydro-silicone oil HSil-100 (RH-H502, hydrogen content of 0.76 wt%, viscosity of 100 mPa·s), and HSil-65 (RH-H33, hydrogen content of 0.18 wt%, viscosity of 65 mPa·s,) were obtained from Zhejiang Runhe Silicone New Materials Co., Ltd., Ningbo, China. Karstedt Pt catalyst (1,3-divinyl-1,1,3,3-tetramethyl-disiloxane-platinum), 5000 ppm (Pt content), was provided by Dongguan Dongsheng Synthetic Materials Co., Ltd., Dongguan, China. ME75 (Proynylol), a catalytic inhibitor, was provided by Momentive, Japan.

### 2.2. Preparation Method of Addition Molding Liquid Silicone Rubber Gel Material

The experimental raw materials were configured into components A and B: the catalyst and vinyl silicone oil were mixed homogeneously in a certain proportion as component A; at the same time, vinyl silicone oil, hydrogen-containing silicone oil, and an inhibitor were mixed homogeneously in a certain proportion as component B. The same amount of components A and B were mixed in a mixer and stirred at a rate of 500 r/min for 30 min to ensure uniform dispersion. Subsequently, the mixture was degassed under vacuum at room temperature for 10 min to remove the air bubbles. Consequently, the mixture was poured into the corresponding mold and vulcanized at 120 °C for 2 h. The formula of the prepared samples is listed in Table 1. According to the -Si-H content (in RH-H502 and RH-H33) and vinyl (-CH=CH_2_) content (in RH-Vi1321), the proportion of Si-H:Si-Vi was set as 0.8, 0.9, 1.0, 1.2, 1.4, and 1.6. The mechanism for the addition reaction of silicone elastomers is shown in Figure 1.

### 2.3. Characterization and Measurements

The compression modulus of the sample was tested using the high-speed rail testing instrument A1-7000S1 (High-speed Rail Technology Co., Ltd., Dongguan, China).

The compression modulus of the silicone rubber gel insulating materials was tested by the test method specified in ASTM D695-15 [12]. According to the standard, the silicone rubber gel insulating material was made into a cylindrical sample with a cross-sectional diameter of 25.4 mm and a height of 12.7 mm. After smearing a thin layer of lubricant on the surface of the polished metal plate, the sample was placed in the center of the compression testing machine to compress the sample at a speed of 1.3 mm/min until the strain reached 25%. The relaxing process for the specimen was carried out at the same speed. The compressing–relaxing process was repeated three more times, and the fourth compression cycle was recorded as the force–deformation curve. Among the four compression cycles, the first three cycles can be considered mechanical adjustments, and the fourth cycle was a formal test. The computer program was set to record the compression modulus value when the strain was 10% and 20% in the fourth compression process for analysis [13].

The breakdown strength test was based on ASTM D149-2013 [14], using an AC medium breakdown tester (Beijing Great Aim Testing Instrument Co., Ltd., Beijing, China), with a boost rate of 0.5 kV s^−1^. A sample with a thickness of about 1 mm was sandwiched between two copper rod electrodes with a diameter of 25 mm and immersed in pure silicone oil to prevent surface flashover.

The breakdown strength of the samples was analyzed by two-parameter Weibull distribution approaches, which reflected the probability of the material being broken down under a certain electric field and the probability of failure after a certain electric field action time, as depicted in the following equation:$$F\left(E,a,\beta \right)=1-exp[-{\left(\frac{E}{\alpha }\right)}^{\beta }]$$
where F(E, a, β) is the cumulative probability density distribution function of the breakdown strength; E is the breakdown strength value obtained from the test; a is the scale parameter, which represents the characteristic field strength of the sample with a cumulative breakdown probability of 63.2%, also known as the Weibull breakdown field strength E; and β is the Weibull shape distribution parameter, which is related to the sample defect and the test conditions [15,16].

For the thermal oxidative aging experiments, the specimens were placed in a 401A thermal-oxidative aging oven (Shanghai Laboratory Instrument Works Co., Ltd., Shanghai, China) at 175 °C for various times (24 h, 48 h, 72 h, 96 h, 120 h, 144 h, and 168 h), and then were kept at room temperature for at least 18 h before further tests.

Fourier transform infrared (FTIR) spectra were collected with a Thermo Scientific Nicolet iS20 (Thermo Fisher Scientific, Waltham, MA, USA) spectrometer equipped with an attenuated total reflectance (ATR) accessory. Nuclear magnetic resonance (NMR) spectra were recorded by a Bruker AVANCE III HD 600 (Bruker, Billerica, MA, USA) instrument with a proton frequency of 600 MHz, in which deuterated chloroform was used as the solvent [17,18,19]. X-ray photoelectron spectra (XPS) were obtained with a Thermo Scientific (Thermo Fisher Scientific, Waltham, MA, USA) K-a X-ray photoelectron spectrometer system. Thermogravimetric analysis (TGA) was performed on a Rigaku TGDTA8122 (Rigaku Corporation, Tokyo, Japan) instrument under nitrogen at a heating rate of 10 °C min^−1^.

## 3. Results and Discussion

### 3.1. Structure of Vinyl Silicone Oil and Methyl Hydro-Silicone Oil

The schematic diagrams for the molecular structure of ViSil-100 and HSil-100 are shown in Figure 2a and Figure 2b, respectively, and the FTIR spectra are shown in Figure 3.

The main characteristic peaks are attributed as follows: The peak at around 2962 cm^−1^ is related to the stretching vibration of the C-H bond in -CH_3_ on Si-Me. The scissor vibration absorption peak of Si-CH_2_- is at around 1410 cm^−1^. The symmetrical deformation and plane swing vibration absorption peaks of Si-Me are at 1261 cm^−1^ and 798 cm^−1^. The peaks at 1087 cm^−1^ and 1018 cm^−1^ are related to the linear Si-O-Si stretching vibration absorption peak [20,21]. The significantly weakened -Si-H peak in HSil-100 at 2158 cm^−1^ proves that most of -Si-H has been involved in the addition reaction, and a small amount of -Si-H is still present in the sample due to the excessive amount of -Si-H in the sample preparation. At 700 cm^−1^–1000 cm^−1^ is the vibrational absorption peak of -CH=CH-. Comparing the infrared spectra of HSil-100 and ViSil-100, the intensity of the absorption peak near 2962 cm^−1^ in the infrared spectrum of ViSil-100 increases, which is due to the superposition of the stretching vibration peak of -CH=CH_2_ and the asymmetric stretching vibration peak of -CH_2_-. The increases in peak intensity near 706 cm^−1^ and 798 cm^−1^ are due to the presence of an out-of-plane deformation absorption peak of -CH=CH_2_. The linear Si-O-Si stretching vibration absorption peak near 1087 cm^−1^ of the cured sample is obvious; compared with the raw material, the absorption peak at 1087 cm^−1^ was more pronounced because the absorption peak at 1018 cm^−1^ is shifted to the left after the curing reaction.

The molecular structures of ViSil-100 and HSil-100 can be further verified by ^1^H NMR and ^13^C NMR spectra, as shown in Figure 2. In testing the NMR spectra, CDCl_3_ was used as a solvent. The ^1^H NMR and ^13^C NMR spectra of vinyl silicone oil ViSil-100 are shown in Figure 4a and Figure 4b, respectively. The triple peak at 6.13 in the ^1^H NMR (600 MHz, CDCl_3_) of ViSil-100 corresponds to hydrogen atom 5 of the above formula, the two sets of multiple peaks at 5.95 and 5.92 correspond to hydrogen atom 4, the multiple peaks at 5.76 and 5.73 correspond to the symmetrical peak of hydrogen atom 4, the peak at 4.43 corresponds to hydrogen atom 1, the peak at 3.73 corresponds to hydrogen atom 8, and the single peak at 3.48 corresponds to hydrogen atom 2. The single peak at 1.55 corresponds to the No. 7 hydrogen atom, and the triple peaks at 1.41 and 1.25 correspond to the No. 3 hydrogen atom and its symmetry peak [22], of which 7.26 is the CDCl_3_ solvent peak. The structure of ViSil-100 was verified by the above spectral analysis. The single peak at ^13^C NMR (151 MHz, CDCl_3_) of ViSil-100 corresponds to carbon atoms 1 and 8 of the above ViSil-100 molecular formula, the single peak at 131.79 corresponds to the hydrogen atoms 2 and 7, and the multiple peaks at 1.20 corresponds to the C atom on the methyl group (-CH_3_), of which 77.16 is the CDCl_3_ solvent peak. The structure of ViSil-100 was further verified by carbon spectroscopy. The ^1^H NMR and ^13^C NMR spectra of the hydrogenated silicone oil HSil-100 are shown in Figure 4c and Figure 4d, respectively. The triple peak at 4.91 in the ^1^H NMR (600 MHz, CDCl_3_) of HSil-100 corresponds to hydrogen atom No. 1 of the above HSil-100 molecular formula, the triple peak at 4.50 corresponds to hydrogen atom No. 5, the triple peak at 4.71 corresponds to the No. 4 hydrogen atom, the triple peaks at 0.18 and 0.11 correspond to the No. 2 hydrogen atom and its symmetry peak, and the single peak at 0.00 corresponds to the No. 3 hydrogen atom, of which 7.26 is the CDCl_3_ solvent peak. The structure of HSil-100 was verified by the above spectral analysis. The multiple peaks at 1.16 in the ^13^C NMR (151 MHz, CDCl_3_) of HSil-100 corresponds to the C atom on the methyl group (-CH_3_), of which 77.16 is the CDCl_3_ solvent peak.

### 3.2. Effect of Viscosity and Content of Methyl Hydro-Silicone Oil on the Properties of Silicone Rubber Gel

The compression modulus data for the silicone rubber gel sample at 20% strain are shown in Figure 5. The ratio of H-Si:Vi-Si is constant, and the higher the viscosity of the hydrogenated silicone oil used in the preparation of silicone rubber gel, the higher the compression modulus of the silicone rubber gel sample. The viscosity of the hydrogen-containing silicone oil is dependent on the silicon–hydrogen content. The higher the molecular weight of hydrogen-containing silicone oils with higher viscosity or higher silicone–hydrogen content, the higher the crosslink density of the silicone rubber gel sample, which will result in a higher compression modulus. Therefore, when HSil-100 is used as the crosslinking agent, the compression modulus is relatively high. For example, when Si-H:Si-Vi = 1.4:1, with ViSil-100 as the matrix and HSil-100 as the crosslinker, the compression modulus is 2.58 MPa, and when HSil-65 is used as the crosslinker, the compression modulus is only 1.36 MPa.

When the ratio of Si-H:Si-Vi changes, the compression modulus of the prepared silicone rubber gel sample is as follows: The compression modulus of the silicone rubber gel sample increases gradually with the increase in the ratio of Si-H:Si-Vi. This is because with the increase in the Si-H:Si-Vi ratio, the crosslinking density of the sample gradually increases, resulting in a gradual increase in the compression modulus. For example, when HSil-100 was used as the crosslinker and ViSil-100 is used as the matrix, the compression modulus increases from 0.60 MPa to 2.90 MPa with the increase in the Si-H:Si-Vi ratio, an increase of 383%. When HSil-65 is used as the crosslinker and ViSil-100 is used as the matrix, the compression modulus increases by 250% from 0.40 MPa to 1.40 MPa with the increase in the Si-H:Si-Vi ratio. It can be concluded that with the increase in the proportion of hydrogen silicone oil in the crosslinker, the crosslinking density increases, and the compression modulus shows an upward trend.

Figure 6 shows the breakdown characteristics of silicone rubber gels with different raw materials and different proportions. With the increase in the proportion of methyl hydro-silicone oil in the crosslinker, the breakdown strength shows an overall upward trend. When HSil-100 is used as a crosslinker, the breakdown of the silicone rubber gel sample gradually increases with the increase in the Si-H:Si-Vi ratio. When HSil-65 is used as a crosslinker, the breakdown strength decreases when Si-H:Si-Vi = 1.6. This is because with the increase in the Si-H:Si-Vi ratio, the crosslinking density of the sample gradually increases, resulting in a gradual increase in the breakdown strength; however, when Si-H:Si-Vi = 1.6, the breakdown strength decreases due to the excessive crosslinking agent content in the sample. For example, when HSil-100 is used as the crosslinker and ViSil-100 is used as the matrix, the breakdown strength increases by 33% from 24.1 kV/mm to 32.2 kV/mm with the increase in the Si-H:Si-Vi ratio. When HSil-65 is used as the crosslinker and ViSil-100 is used as the matrix, the breakdown strength increases from 22.2 kV/mm to 29.1 kV/mm with the increase in the Si-H:Si-Vi ratio, which is 31%. When Si-H:Si-Vi = 1.4, the breakdown strength reaches the highest value, and when Si-H:Si-Vi = 1.6, the breakdown strength decreases to 23.1 kV/mm.

### 3.3. Effect of Aging Time on the Properties of Silicone Rubber Gel

The thermal-oxidative-aged silicone rubber gel samples were prepared using HSil-100 as crosslinker and ViSil-100 as the matrix, and the Si-H:Si-Vi = 1.4:1 samples were aged (thermal-oxidative aging) at an aging temperature of 175 °C. The high elasticity of polymer materials is due to the presence of many crosslinked junctions along the polymer backbone [23]. The increase in the crosslinking density will inhibit sliding between the polymer chains reduce the mobility of the polymer, and lead to an increase of polymer hardness and tensile strength [24]. The compression modulus of the aged sample is shown in Figure 7, and it can be seen that the compression modulus increases with the increase in aging time, and the compression modulus increases by 49.2% after 168 h. This is because, with the increase in thermal-oxidative aging time, the crosslinking density of the liquid silicone rubber gel gradually increases, leading to a gradual increase in the compression modulus.

Temperature affects the breakdown strength of polymer materials, and when the temperature is lower than the glass transition temperature of the polymer, the breakdown strength is independent of temperature, and, vice versa, the breakdown strength decreases with increasing temperature. Silicone rubber gels have a relatively low glass transition temperature; the glass transition temperature of silicone rubber is about −50 °C [25]. The temperature range of the aging test is much higher than its glass transition temperature, so the breakdown strength is inversely correlated with temperature.

As can be seen from Figure 8, the breakdown strength decreases with the increase in aging time, and the breakdown voltage strength of the silicone rubber gel sample decreases by about 30% after 168 h of aging. The AC breakdown strength of the silica rubber gel decreases with the aging time, which is due to the volume expansion and thermal deformation of the silicone rubber gel during thermal aging. The sample shrinks due to thermal expansion and contraction, and the internal defects increase after aging, which is more conducive to the accelerated migration of the free electrons in the material, and the collision ionization between the electrons is more likely to occur.

### 3.4. Structure of Silicone Rubber Gels before and after Aging

The FITR spectra of the samples before and after aging are shown in Figure 9. The main characteristic peaks are attributed as follows: The vicinity of 2962 cm^−1^ and 2895 cm^−1^ are the stretching vibration absorption peaks of C-H bonds of -CH_3_ and -CH_2_- on Si-Me. The scissor vibration absorption peak of Si-CH_2_- is near 1410 cm^−1^. The symmetrical deformation and plane swing vibration absorption peaks of Si-Me are near 1261 cm^−1^ and 798 cm^−1^. The vicinity of 1087 cm^−1^ and 1018 cm^−1^ is the linear Si-O-Si telescopic vibration absorption peak [26]. Compared with the spectrum of HSil-100, the intensity of the peak near 2158 cm^−1^ in Figure 9 is greatly weakened, which proves that most of -Si-H is involved in the addition reaction, and a small amount of -Si-H is still present in the sample due to the excess of -Si-H during the preparation of the sample. Comparing the FTIR spectra of the samples before and after aging, it can be seen that the absorption peak near 2962 cm^−1^ in the spectra is almost the same, which is different from the infrared spectra of silicone oil mentioned above because -CH=CH_2_ is involved in the reaction, and when the sample is prepared, there is an excess of -Si-H, so there is almost no peak characteristics near -CH=CH_2_ in the sample, and the same is true for the peak near 798 cm^−1^. The infrared spectra of the samples before and after aging were almost identical, indicating that the structural characteristics of the samples change very little after aging, which indicated that the silicone rubber gel samples prepared in this work had excellent aging resistance.

The XPS spectra of the sample before and after aging are shown in Figure 10. Combining the correlation peaks in Figure 10 and XPS spectra, the characteristic peak of Si2p is 102.33 eV, the characteristic peak of Si2s is 155 eV, the characteristic peak of C1s is 284.8 eV, and the characteristic peak of O1s is 532.42 eV. Table 2 shows the comparison of the elemental content of silicone rubber gel samples before and after aging, which was obtained by XPS spectrum analysis. It can be seen from the table that the element content of the samples before and after aging is almost the same, indicating that the aging does not change the structural characteristics of the sample, which is consistent with the infrared spectrogram results, indicating that the silicone rubber gel sample has excellent aging resistance.

The TGA curves of the samples before and after aging are shown in Figure 11. As can be seen from the figure, the TGA temperature range is 40–800 °C. The thermal decomposition of the sample before aging can be divided into two stages: in the range of 350–600 °C, the mass loss of the product is 56.4%; there is a 19.2% loss in mass between 600 and 680 °C. The final residue mass at 800 °C is 24.4%. In the range of 350–600 °C, the front section of the weight loss curve is gentle, and it begins to change abruptly at about 380 °C, indicating that the sample begins to degrade thermally, and this stage is mainly the thermal decomposition of methyl groups [27,28]. Most of the organic compounds begin to oxidize and decompose at 250–300 °C, and the thermal decomposition temperature of the silicone rubber gel prepared in this report is high, which proves that the sample has good aging resistance. The mass loss between 600 and 680 °C is mainly due to the breakage and decomposition of the silicon–oxygen (-Si-O-) backbone. Above 700 °C, the thermal decomposition of the sample was completed, the mass gradually stabilized, and the residual amount was 24.4%; the possibility of some residual carbon was reduced at the end of 800 °C. The thermal decomposition of the aged sample is also divided into two stages: in the range of 350–625 °C, the mass loss percentage of the product is 49.8%; the percentage loss in mass between 625 and 750 °C is 17.85%. The final residue mass is 32.35% at 800 °C. The residual mass of the aged sample is higher than that of the unaged sample due to the decomposition of some of the methyl groups during the aging process.

## 4. Conclusions

Through the above experimental exploration, the following conclusions can be drawn: The higher the content of methyl hydro-silicone oil, the higher the crosslinking density of the sample, and the breakdown strength and compression modulus increase with the increase in the methyl hydro-silicone oil content. When Si-H:Si-Vi = 1.4:1, the performance is the best. With the increase in aging time, the compression modulus showed an upward trend, and the breakdown strength showed a downward trend. After 168 h, the compression modulus increased by 49.2%, and the breakdown voltage strength decreased by about 30.0%. With the increase in thermal-oxidative aging time, the cross-linking density of the liquid silicone rubber gel gradually increases, and the compression modulus gradually increases. Due to the increase in sample defects, the breakdown strength gradually decreases, and the aging resistance of the silicone rubber gel samples is better.

## Figures and Tables

**Figure 1 polymers-16-00763-f001:**
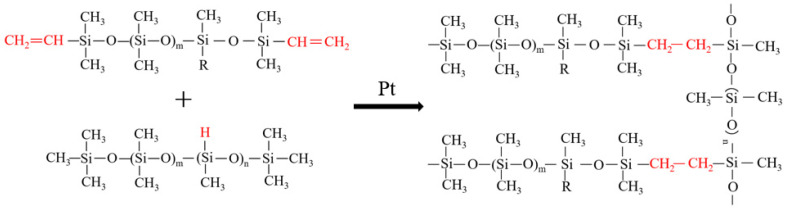
Schematic diagram of the addition reaction of silicone elastomers.

**Figure 2 polymers-16-00763-f002:**
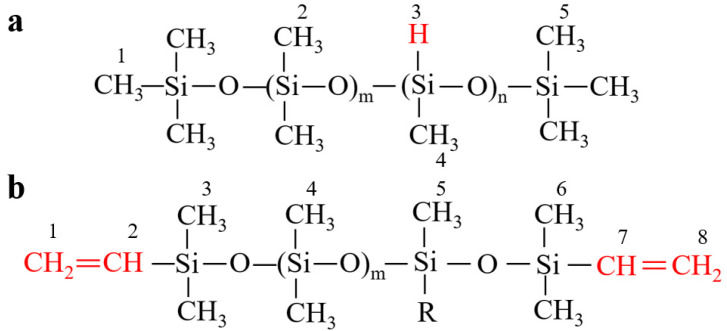
Schematic diagram of the molecular structure of (**a**) HSil-100 and (**b**) ViHSil-100.

**Figure 3 polymers-16-00763-f003:**
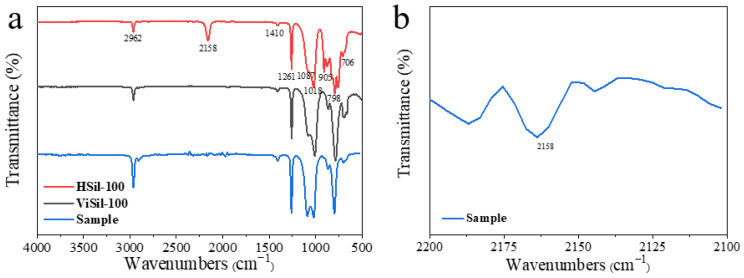
(**a**) FTIR spectra of ViSil-100, HSil-100, and cured samples; (**b**) FTIR spectra of cured samples in the range 2100–2200 cm^−1^.

**Figure 4 polymers-16-00763-f004:**
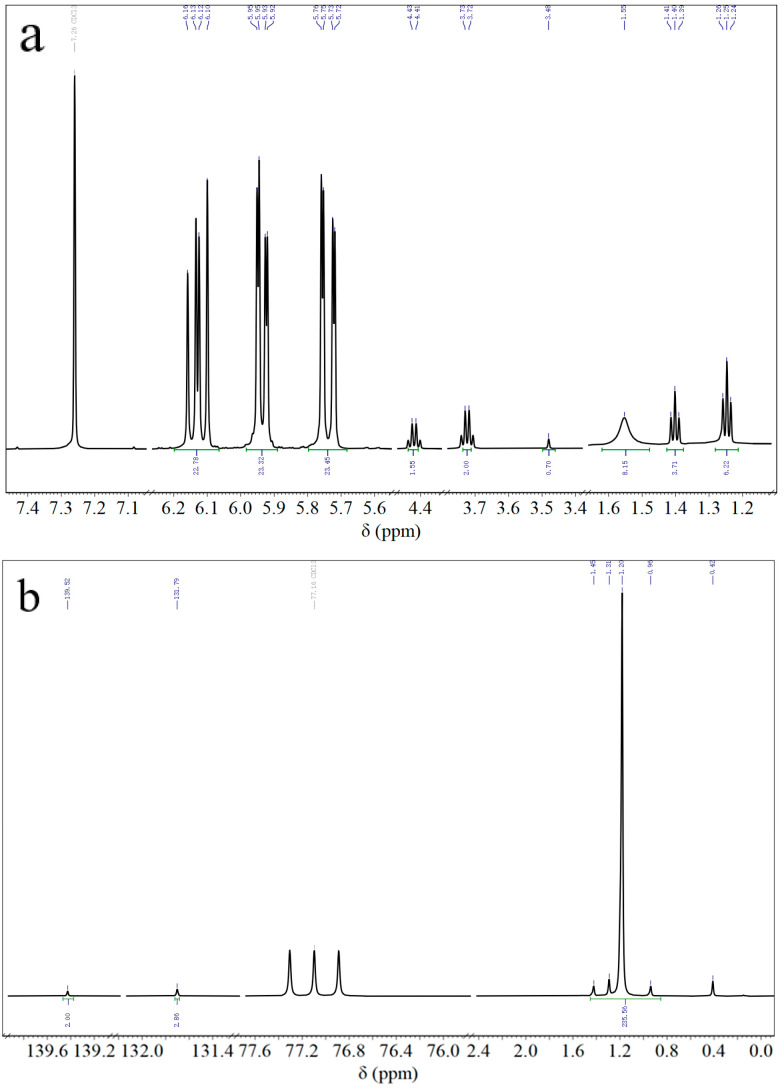
(**a**) ^1^H NMR spectra of ViSil-100; (**b**) ^13^C NMR spectra of ViSil-100; (**c**) ^1^H NMR spectra of HSil-100; (**d**) ^13^C NMR spectra of HSil-100.

**Figure 5 polymers-16-00763-f005:**
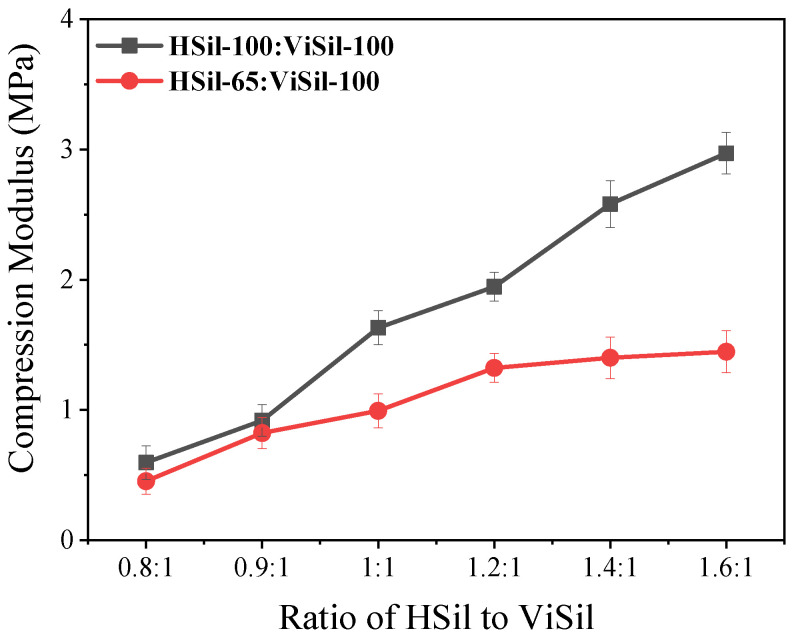
Schematic diagram of the compression modulus of silicone rubber gels with different hydrogen-containing silicone oils as crosslinkers.

**Figure 6 polymers-16-00763-f006:**
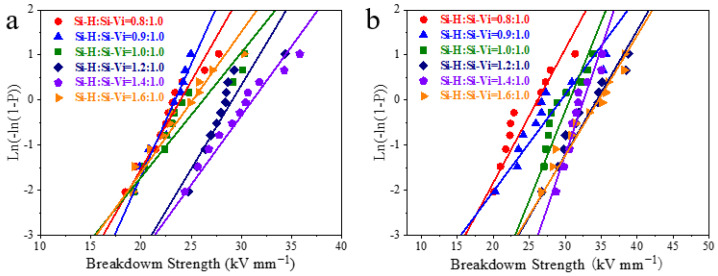
Weibull distribution of the breakdown strength of silicone rubber gels: (**a**) ViSil-100 as the matrix and HSil-65 as the crosslinker; (**b**) ViSil-100 as the matrix and HSil-100 as the crosslinker.

**Figure 7 polymers-16-00763-f007:**
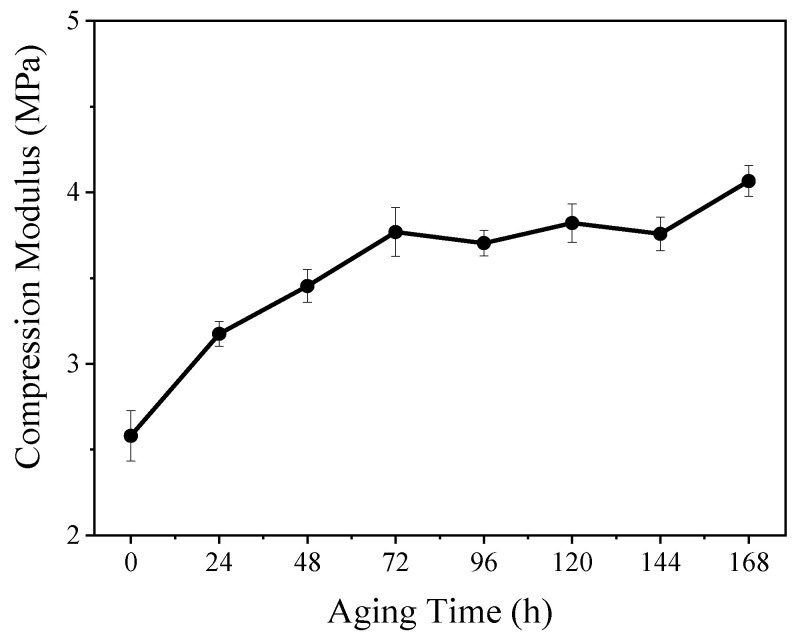
Compression modulus of silicone rubber gel at different aging times.

**Figure 8 polymers-16-00763-f008:**
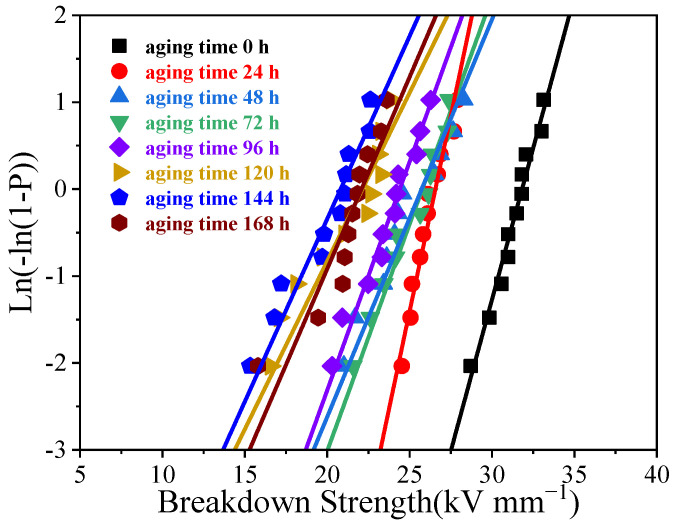
Weibull distribution of breakdown strength of silicone rubber gel at different aging times.

**Figure 9 polymers-16-00763-f009:**
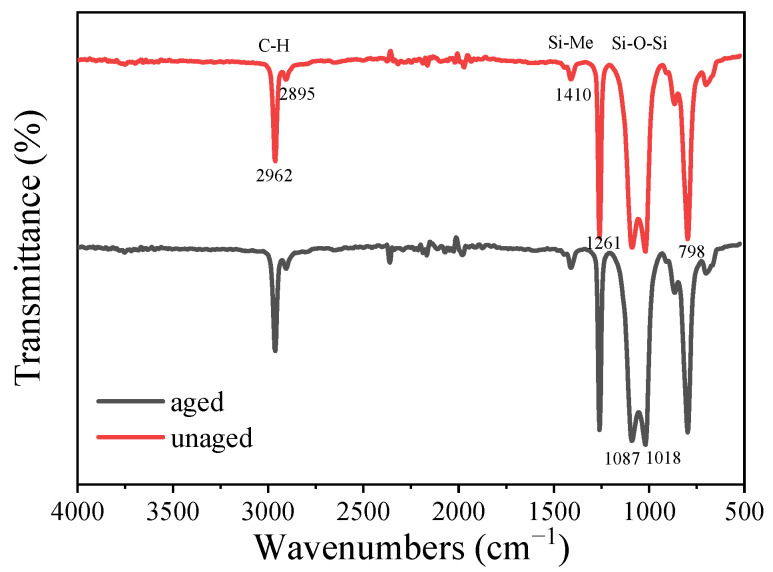
FTIR spectra of silicone rubber gel samples before and after aging.

**Figure 10 polymers-16-00763-f010:**
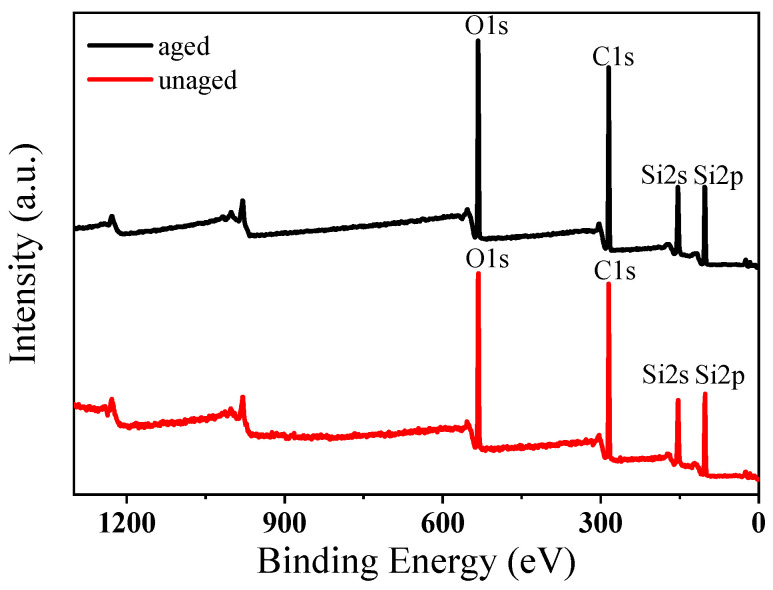
XPS spectra of the sample before and after aging.

**Figure 11 polymers-16-00763-f011:**
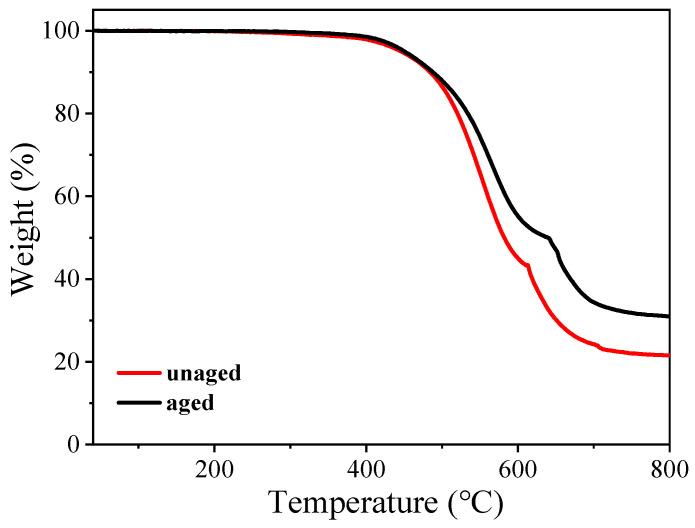
TGA curves of the sample before and after aging.

**Table 1 polymers-16-00763-t001:** Formula of silicone rubber gels.

Samples	ViSil-100 (g)	HSil-65 (g)	HSil-100 (g)	Pt catalyst (g)	ME75 (g)
0.8:1	100	17.62	0	0.02	0.04
0.9:1	100	19.82	0	0.02	0.04
1.0:1	100	22.02	0	0.02	0.04
1.2:1	100	26.42	0	0.02	0.04
1.4:1	100	30.81	0	0.02	0.04
1.6:1	100	35.23	0	0.02	0.04
0.8:1	100	0	4.18	0.02	0.04
0.9:1	100	0	4.70	0.02	0.04
1.0:1	100	0	5.22	0.02	0.04
1.2:1	100	0	6.26	0.02	0.04
1.4:1	100	0	7.31	0.02	0.04
1.6:1	100	0	8.35	0.02	0.04

**Table 2 polymers-16-00763-t002:** Comparison table of elemental content of samples before and after aging.

	Unaged			Aged		
	Peak BE	FWHM eV	Atomic %	Peak BE	FWHM eV	Atomic %
Si2p	102.33	1.36	25.9	102.36	1.36	25.69
C1s	284.8	1.2	49.37	284.81	1.18	48.82
O1s	532.42	1.22	24.73	532.43	1.2	25.49

## Data Availability

Data are contained within the article.

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
