# Peer review of "Effect of Methyl Hydro-Silicone Oil Content and Aging Time on Compression Modulus and Breakdown Strength of Additional Liquid Silicone Rubber Gel"

_polymers, 2024, doi:10.3390/polym16060763_

Round 1

Reviewer 1 Report

Comments and Suggestions for Authors

This manuscript describes the fabrication of silicon rubber gels and evaluation of the effect of methyl hydrosilicone oil and aging time. The formed silicon rubber gels are characterized by IR and NMR measurements, and their mechanical properties were characterized by compression tests. The results are scientifically described and discussed based on scientific evidence. This manuscript contains valuable results in certain contents, however, I feel the major revisions are required to be published in in Polymers. My comments are below.

1.     Most importantly, what are the novelty and scientific finding in this manuscript? I understood the authors evaluated the effects of crosslinking density and aging time of silicon rubber gels. However, there are various types of silicon rubbers, which were already reported. Comparing these previous reports, please describe the difference and findings of this manuscript appropriately.

2.     Page 5, line 150. “,,, a small amount of -Si-H is still presented in the sample due to the excessive amount of -Si-H in the sample preparation.” This reviewer can not find the remaining Si-H peak. To clarify this point, please make the figure with enlarged view around 2100 to 2200 cm–1.

3.     Page 5, line 157. “The linear Si-O-Si stretching vibration absorption peak near 1087 cm–1 of the cured sample is obvious, which is due to the characteristic absorption peak generated by the curing sample.” Why does this peak appeared so strongly after curing? In my opinion, the mobility of Si-O-Si bond does not change so much before and after curing (or might be more hard to move because of the formation of crosslinking points, which would be made it hard to find the peak of this stretching.)

4.     Page 7, line 206. “when HSil-100 is used for hydrogenated silicone oil, the compression modulus of the silicone rubber gel sample increases gradually with the increase of the ratio of Si-H: Si-Vi, and when HSil-65 is used for hydrogenated silicone oil, the compression modulus of the silicone rubber gel sample increases gradually with the increase of the Si-H: Si-Vi ratio”. The sentences are duplicated. Please makes it more simply in one sentence.

5.     Page 7, line 210. “when the HSil: ViSil ratio continues to increase, the compression modulus of the partial ratio decreases.” I do not see the decrease of the compression modulus with increasing the ratio in Figure 5.

6.     Page 7, line 215. “For example, when HSil-100 was used as the  crosslinker and ViSil-100 was used as the matrix, the compression modulus increased from 2.12 MPa to 11.98 MPa with the increase of the Si-H: Si-Vi ratio, an increase of 465%. When HSil-65 was used as the crosslinker and ViSil-100 was used as the matrix, the compression modulus increased by 192% from 1.89 MPa to 5.51 MPa with the increase of the Si-H: Si-Vi ratio.” These values do not corresponded to the values described in Figure 5.

7.     Page 9, line 260. “Silicone rubber gels have a glass transition temperature of 120 °C,,,” Was this value determined by DSC measurements? If so, how about the values with after Si-H: Si-Vi ratios? I thought this value is relatively high for silcon rubbers.

8.     Page 10, line 280. “Compared with the spectrum of HSil-100, the intensity of the peak near 2158 cm-1 in Figure 9 is greatly weakened,,,” Related to comment #2, I can not find the peak around 2158 cm–1. Please makes figure with enlarged view.

9.     Figure 9. Can you see the peaks corresponded to C=C bond in C1s region of XPS?

Reviewer 2 Report

Comments and Suggestions for Authors

The paper entitled "Effect of methyl hydro-silicone oil content and aging time on compression modulus and breakdown strength of additional liquid silicone rubber gel" reports on the preparation and evaluation of properties of two-component silicone gel materials.

First of all, I want to point out that the novelty of this paper is missing, such materials and characterization techniques are already prepared and reported.

The title is not suggestive for the content of this paper. It is not clear what is the "additional liquid silicone rubber gel" .

The abstract needs also improvements. The first paragraph must be moved to the introduction and a reference is needed for the statements.

It is not clear the aging time noted in Abstract.

Introduction - too general information and already known about the hydrosilylation reaction in the presence of the Pt catalysts. At line 47 is not clear if LA is an abbreviation. At line 53 a reference is needed.

The motivation of this study must be highlighted, the advantages of the silicone rubber gels, etc. The content of Introduction is poor and must be changed.

Section 2.1. line 74 - what Pt catalyst was used?

Section 2.2. Figure 1 does not respect the structure of the monomer, especially of the hydrogen silicone oil, so the structure of the product is not well represented. In Table 1 it is not clear how the ratio was calculated?

In Table 1 the concentration of the Pt catalyst is in ppm, while of the inhibitor is in g. Somehow, seems unreal.

Section 3. Fig. 4 is unclear, the X axis is unclear, also the peak position, the integration must be presented. In fig. 3(d) X-axis can't be for the 13C NMR. The solvent must be mentioned. I suggest that the spectra to be presented separately in the supplementary.

Fig 5. How authors explain the increase of the compression modulus based on the crosslinker concentrations?

In this section I recommend that some comparisons with other reports to be added.

Thermal analysis must be also be discussed by means of DSC analysis.

Based on these observations I consider that this paper needs general revision, so it can not be published as such.

Round 2

Reviewer 1 Report

Comments and Suggestions for Authors

The authors have made the revisions according to the reviewer’s comments. I appreciate their efforts, however, I still feel some points should be clarified before the acceptance. My comments are below.

#1. I do understand what the authors have done in this manuscript. The authors gave the comments that “we talk about the influence of the viscosity of methyl hydro-silicone oil (the hydrogen content of the methyl hydro-silicone oil) on the properties of the addition of liquid silicone rubber”. However, the authors only examined two methyl hydro-silicone oil. So, it is kind of difficult to give systematic finding at this point. The authors also do not compare these two methyl hydro-silicone oil and make the conclusion about the effects of them.

#2. Thank you for making enlarged IR spectra. The resolution of inserted Figure is quite low and therefore hard to see it. Please increase the resolution of figure.

#3. The authors gave the comments that “Compared with the raw material, the peak at 1087 cm-1 was more obvious because the Si-O-Si content per unit volume increased after the reaction of the raw material”. However, I do not agree the author’s comment because both backbone polymer (ViSil) and crosslinker (HSil) contain Si-O-Si bonds in almost similar content. Although they have different functional groups (Si-H and Si-CH=CH2), these contents are very small in the polymer chains and they mainly contain Si-O-Si repeating unit. Therefore, the content of Si-O-Si per unit volume should not change so much.

#7. Please add the reference that describe the glass transition temperature of common silicon rubbers. I think there are many related papers.

#8. I understand Si-H bond is almost disappeared after curing. So, could the authors see the difference before and after reaction? My concern is that the Si-H bond of unaged sample is originally not so obvious. But if this small peak is disappeared after reaction, the authors can strongly say the peak of unaged sample is really Si-H bond.

#9. Why C=C bond peak is not found in XPS although this peak was found in IR? Please add some description in the manuscript.

#Additional comment1. How does the authors determine the Si-H : SiVi ratio? Is this the molar ratio based on their functional groups or the weight ratio of materials? Please add some description how the authors define it. According to Table 1, it would be the molar ratio. If my opinion is correct, at least twice of Si-H is needed to SiVi. However, the authors mentioned “when HSil: ViSil=1.6, the breakdown strength decreases due to the excessive crosslinking agent content in the sample” in page 7, line 232. The crosslinking agent should not be excess in this content in my understanding. Additionally, to clarify this tendency, further experiments such as HSil: ViSil = 1.8 or 2.0 should be done.

#Additional comment2. The authors use the description “Si-H : SiVi” and “HSil: ViSil” to say the ratio of two silicon oil. Please unify the description thorough the manuscript.

Reviewer 2 Report

Comments and Suggestions for Authors

The authors responded to most of the suggestions made but some have remained unresolved, so I argue that the paper should be revised especially in terms of the quality of the figures:

Fig.1 - the authors started from a bifunctional vinyl-silicone compound and a monofunctional H-silicone, so the result of this reaction is not the product presented by the authors, so the figure must be revised.

Fig. 3 - The peaks are not visible especially in the inset graph. I suggest that the inset graph to be represented as Fig.2 b

Fig. 4- the authors claimed in their responses that the Fig. 3 was improved, but is unchanged in the revised manuscript. I suggest that separate figures to be presented in the revised manuscript, the scale to be visible, the integrals to be added.

Round 3

Reviewer 1 Report

Comments and Suggestions for Authors

I'm not convinced with author's reply in some points. However, I also feel the manuscript reaches the enough quality of the journal.

Reviewer 2 Report

Comments and Suggestions for Authors

The paper was revised and can be accepted.